# Flow Cytometric Analysis of Oxidative Stress in *Escherichia coli* B Strains Deficient in Genes of the Antioxidant Defence

**DOI:** 10.3390/ijms23126537

**Published:** 2022-06-10

**Authors:** Beatriz Jávega, Guadalupe Herrera, José-Enrique O’Connor

**Affiliations:** 1Laboratory of Cytomics, Joint Research Unit CIPF-UVEG, University of Valencia, 46010 Valencia, Spain; beatriz.javega@uv.es; 2Flow Cytometry Unit, IIS INCLIVA, Fundación Investigación Hospital Clínico Valencia, 46010 Valencia, Spain; gherreramartin@gmail.com

**Keywords:** superoxide dismutase, catalase, OxyR, SodA, SodB, organic hydroperoxides, hydrogen peroxide, fluorescence

## Abstract

The detection of reactive oxygen species (ROS) and the analysis of oxidative stress are frequent applications of functional flow cytometry. Identifying and quantifying the ROS species generated during oxidative stress are crucial steps for the investigation of molecular mechanisms underlying stress responses. Currently, there is a wide availability of fluorogenic substrates for such purposes, but limitations in their specificity and sensitivity may affect the accuracy of the analysis. The aim of our work was to validate a new experimental model based in different strains of *Escherichia coli* B deficient in key genes for antioxidant defense, namely *oxyR*, *sodA* and s*odB*. We applied this model to systematically assess issues of specificity in fluorescent probes and the involvement of different ROS in a bacterial model of oxidative stress, as the probes can react with a variety of oxidants and free radical species. Our results confirm the higher sensitivity and specificity of the fluorescent probe mitochondrial peroxy yellow 1 (MitoPY1) for the detection of H_2_O_2_, and its very low capacity for organic hydroperoxides, thus extending MitoPY1′s specificity for H_2_O_2_ in mammalian cells to a bacterial model. On the contrary, the fluorescent probe 2′,7′-dichlorodihydrofluorescein diacetate (H_2_DCF-DA) is more sensitive to organic peroxides than to H_2_O_2_, confirming the lack of selectivity of H_2_DCF-DA to H_2_O_2_. Treatment with organic peroxides and H_2_O_2_ suggests a superoxide-independent oxidation of the fluorescent probe Hydroethidine (HE). We found a positive correlation between the lipophilicity of the peroxides and their toxicity to *E. coli*, suggesting greater quantitative importance of the peroxidative effects on the bacterial membrane and/or greater efficiency of the protection systems against the intracellular effects of H_2_O_2_ than against the membrane oxidative stress induced by organic peroxides. Altogether, our results may aid in preventing or minimizing experimental errors and providing recommendations for the proper design of cytometric studies of oxidative stress, in accordance with current recommendations and guidelines.

## 1. Introduction

Oxidative stress is involved in cell senescence, aging and in many diverse genetic and acquired conditions [1]. Oxidative stress may be initiated and amplified by a variety of oxygen (O_2_) reactive species (ROS), mostly free radicals, i.e., highly reactive chemicals with unpaired electrons [2]. 

Biologically relevant ROS derived from O_2_ metabolism include superoxide anion radicals (O_2_^−^), hydrogen peroxide (H_2_O_2_) and hydroxyl radicals (OH^.^) [1]. O_2_^−^ is a relatively unreactive species that can interact with other molecules to generate H_2_O_2_ and other ROS through enzyme- or metal-catalyzed processes [3]. H_2_O_2_ is not a free radical, as it lacks unpaired electrons, but diffuses easily through membranes and has a relatively long half-life [4], allowing H_2_O_2_ to undergo metal-catalyzed reactions that yield the OH^.^ radical. OH^.^ is one of the strongest free radicals and reacts readily with cellular components, including DNA, proteins, lipids and carbohydrates, thus being one of the most potentially damaging ROS [2]. 

On the other hand, ROS also serve important regulatory roles, mediated by intercellular and intracellular signaling, including the adaptation to endogenous and exogenous stress and the destruction of invading pathogens [5].

The role of ROS in bacterial metabolism and stress responses is a very important issue in physiological and pathological processes. ROS are unavoidable byproducts of O_2_ exposure and utilization by bacterial cells [6]. ROS generation under aerobic conditions may induce oxidative stress in bacteria and damage multiple cellular targets, including iron-sulfur clusters, cysteine and methionine protein residues, and DNA [7].

The gene-dependent regulatory responses to O_2_^−^ and H_2_O_2_ in *E. coli* are mostly mediated by the induction of superoxide dismutases (SOD) and catalases, respectively. *E. coli* has three SODs, MnSOD (sodA), FeSOD (sodB) and CuZnSOD (sodC), that dismutate O_2_^−^ to H_2_O_2_. Catalases, belonging to the peroxidase family of enzymes, further degrade H_2_O_2_ to H_2_O and O_2_ [6]. *E. coli* has two catalases, hydroperoxidase I (HPI) and hydroperoxidase II (HPII), encoded by *katG* and *katE* [8,9]. The SOD and catalase genes of *E. coli* are members of two important regulators of oxidative stress, the OxyR and the SoxRS regulons, respectively [10].

Flow cytometry has become a choice method for bacterial research. In bacterial models, fluorescent probes are used in attempts to define the role of ROS production [11,12,13,14]. However, the detection of ROS is a complex task due to the low concentration, short half-life, and extensive interactions of ROS, as well as the intrinsic limitations of both probes and experimental conditions [15]. In addition, the efficiency and specificity of different dyes for detecting ROS in vitro have rarely been established with any confidence [15], much less in bacteria. Such limitations and potential sources of artifacts cause the quantitative measurement of the intracellular generation of ROS to be a difficult challenge, and solving this issue requires the careful design of experiments and the cautious interpretation of the results [16]. 

While using bacteria to study oxidative stress may be advantageous, functional cytometric assays in live bacteria are still limited. This is due mostly to the cell wall impairing penetration of vital dyes in bacteria, thus imposing permeabilization procedures. These time-consuming manipulations affect cell physiology and may result in aggregation or cell lysis. In previous studies by our laboratory [11,12], we assessed the application of *E. coli* B strain as an alternative to permeabilization steps in flow cytometry functional assays. *E. coli* B has been and is still extensively used for mutagenic assays [17,18,19,20], while the *E. coli* K12 strain is applied mostly to genetic and biochemical studies. Multi-omic analysis confirmed the different cell-wall and outer-membrane composition of *E.coli* B, and predicted this strain to be more favorable for both protein secretion and the uptake of exogenous chemicals [21]. In accordance with this, our previous work showed that the *E. coli* B strain IC188 exhibited more efficient staining with vital fluorochromes than the *E. coli* K-12 strain AB1157 while maintaining a similar membrane potential. In addition, we found the IC188 strain to be more sensitive than AB1157 for revealing oxidative stress when challenged with prooxidants, supporting its suitability as a biosensor of oxidative stress [11,12].

In order to extend our previous findings, and to provide a suitable bacterial model for the accurate detection of ROS, we characterized two novel biosensors derived from the *E. coli* B parental wild type (strain IC188) which are deficient either in the OxyR function (strain IC203) or simultaneously in OxyR, SodA and SodB functions (strain IC5233). Based on these models, we quantified the intracellular levels of ROS by means of flow cytometry using ROS-sensitive fluorescent probes after exposure to three relevant peroxidative xenobiotics, namely hydrogen peroxide (H_2_O_2_), tert-butyl hydroperoxide (t-BOOH) and cumene hydroperoxide (CHP), differing in solubility and prooxidant mechanisms [20].

## 2. Results

### 2.1. Assessment of the Sensitivity of the Bacterial Strains to Peroxidative Xenobiotics

Table 1 compares the sensitivity to oxidants of the IC188 wild-type strain and the two mutant strains. The OxyR regulon is the transcriptional regulator in *E. coli* for the expression of antioxidant genes in response to oxidative stress, in particular to increased levels of H_2_O_2_ [10]. The SoxRS regulon controls *sodA* and *sodB* genes and acts as a redox sensor of O_2_^−^. [10]. As expected, in concordance with their gene deficiencies, strains IC203 and IC5233 exhibited higher sensitivity to exogenous prooxidants, with some differences between them. IC203, deficient in OxyR, was sensitive to all prooxidants tested, most clearly to t-BOOH. The triple mutant strain IC5233, deficient in OxyR, SodA and SodB, was curiously less sensitive to t-BOOH, but was the most sensitive to menadione, a compound generating O_2_^−^ by redox cycling [22].

### 2.2. Identification and Gating of Single Live Bacteria for Flow Cytometric Analysis

Setting up a flow cytometric analysis of cell function requires the clear identification of single live cells in order to exclude both dead or dying cells and cell doublets [13,23]. While these procedures are relatively easy to perform when analyzing human or mammalian cells, they become complicated for bacteria, mostly due to their smaller size. 

Figure 1 shows the strategy we followed to identify single live bacterial cells, separated from debris and cell doublets (or aggregates). The scatter dotplot in panel 1 (FS integral vs. SS integral) shows two populations of events, likely including bacteria and debris. Because of the small size of bacteria, scatter signals are amplified and registered on a logarithmic scale [12]. Panel 2 shows the dotplot allowing us to discriminate single cells from doublets/aggregates (FS peak vs. FS integral, logarithmic scale). The identification of live bacteria in panel 1 is confirmed with the probe SYTO-9, which stains live cells (panel 3) and with PI, which stains only dead or dying cells (panel 4). The population identified as live bacteria in panel 1 is the only one containing events able to take up SYTO-9 (panel 3) and, upon toxic exposure, take up PI and lose the morphological features of live bacteria (panel 4). The population identified as debris in panel 1 is unable to take up either dye. Accordingly, all of the analysis related to the use of ROS-sensitive fluorescent probes in our study was performed on events that were included simultaneously in the gate of “live bacteria” (panel 1) and “single cells” (panel 2). 

Following these cytometric criteria, we could obtain meaningful results related to intracellular ROS determination, as exemplified in panel 5, which shows the dose-dependent intracellular fluorescence of the probe H_2_DCF-DA in samples exposed to growing concentrations of t-BOOH. This example also shows clear differences between the control strain IC188 (A) and the strain IC203 (B), which is deficient in oxyR, a function essential for antioxidant defense.

### 2.3. Titration of ROS-Sensitive Fluorescent Probes in Bacteria

A very important procedure when setting up a florescence-based determination in flow cytometry is to titrate the fluorescent probes properly, in order to define their optimal staining concentration for minimizing saturation and the suboptimal detection of the relevant biological parameters [23].

The titration of green-emitting fluorescent reagents H_2_DCF-DA and MitoPY1 (Figure 2) was performed with the control strain IC188, using t-BOOH to generate intracellular peroxides, as both fluorescent probes have been previously described as peroxide-sensitive [16]. Unstained suspensions of strain IC188 exposed to 10 mM t-BOOH showed a fluorescence ratio of 1.05 ± 0.02 with respect to untreated samples. H_2_DCF-DA concentrations ranging from 2.05 µM to 41.04 µM resulted in a concentration-dependent increase in the fluorescence ratio. Increasing concentrations resulted in a decreased ratio. Therefore, 41.04 μM was selected to analyze oxidative stress in *E. coli* with H_2_DCF-DA (Figure 2A).

For MitoPY1 titration, the baseline ratio was 1.29 ± 0.14. With increasing concentrations of MitoPY1 up to 5 µM, a concentration-dependent increase in fluorescence intensity was observed. Increasing concentrations resulted in decreased ratio. Therefore, 5 μM was selected to analyze oxidative stress in *E. coli* with MitoPY1 (Figure 2B).

The titration of the orange-emitting fluorescent probe HE (Figure 3) was performed on the triple mutant strain IC5233, using menadione to generate intracellular O_2_^−^, as this fluorescent dye has been described as most sensitive to this ROS [16]. Unstained bacterial suspensions exposed to 0.29 mM menadione showed a fluorescence ratio of 1.08 ± 0.03 with respect to untreated samples. After testing a range of HE concentrations up to 25 µM, the optimal concentration of HE was found to be 15.88 µM, as fluorescence appeared to be saturating at a higher concentration.

### 2.4. Effect of Prooxidant Treatment on Viability and Intracellular Oxidative Stress in the Different Bacterial Strains

#### 2.4.1. Effect of H_2_O_2_ Treatment

Figure 4 shows the results of the viability assay to determine the cytotoxicity of H_2_O_2_ on the three *E. coli* strains, IC188, IC203 and IC5233. H_2_O_2_ induced concentration-dependent cell death in all strains, with the wild-type strain IC188 being the least sensitive (IC50 = 1342 mM). As expected, in view of its deficiency in *oxyR*, *sodA* and *sodB* genes, the strain IC5233 was the most sensitive strain to H_2_O_2_ (IC50 = 847 mM). The oxyR mutant IC203 strain showed slightly higher sensitivity to H_2_O_2_ than the wild-type strain, especially at the highest H_2_O_2_ concentrations tested. However, the overall IC50 of IC203 (1430 mM) was similar to that of IC188 strain, due to the lower intrinsic viability of IC203 in aerobic conditions, as previously described [18].

From these IC_50_ curves, it is clear that H_2_O_2_ concentrations lower than 100 mM resulted in less than 15% lethality in any strain. Accordingly, the range of H_2_O_2_ concentrations used to determine intracellular ROS was 0–100 mM. Under these experimental conditions, the generation of ROS secondary to cell death processes could be minimized. Moreover, the analysis of intracellular ROS was restricted to living cells, based on the gating strategy shown in Figure 1. 

The quantification of intracellular ROS in the wild-type strain IC188 and the strains deficient in *oxyR* (strain IC203) or in *oxyR*, *sodA* and *sodB* (strain IC5233) exposed to exogenous H_2_O_2_ is shown in Figure 5.

As seen in Figure 5, in all strains, both green fluorescent probes H2DCF-DA (Figure 5A) and MitoPY1 (Figure 5B) could detect H_2_O_2_, a hydrosoluble peroxide. However, MitoPY1 showed greater sensitivity to H_2_O_2_ than H_2_DCF-DA. MitoPY1 was also superior in revealing differences among the strains, with the triple mutant IC5233 strain being the most sensitive to exogenous H_2_O_2_. 

In another series of experiments, we used the dye HE to provide some insights into the specificity of fluorescent probes and the involvement of different ROS in the action of the exogenous peroxides tested. HE has been widely used to detect O_2_^−^ radicals based on its reported specificity for this ROS [24]. The orange fluorescence emission of HE when oxidized was also an interesting point, as it could be combined with the green-emitting fluorochromes used in the study.

As shown in Figure 5C, exposure to exogenous H_2_O_2_ induced a dose-dependent increase in the HE fluorescence ratio in all strains, without significant differences among wild-type and mutant strains. The increase in the HE fluorescence ratio was of slightly higher intensity than that observed for DCF fluorescence, but much lower than that seen for the MitoPY1 probe under the same experimental conditions.

These results suggest that MitoPY1 may be the choice fluorescent probe for detecting intracellular H_2_O_2_ in bacteria and that intracellular O_2_^−^ is generated to some extent in *E. coli* cells exposed to exogenous H_2_O_2_ [25].

#### 2.4.2. Effect of t-BOOH Treatment

Figure 6 shows the results of the viability assay to determine the cytotoxicity of t-BOOH on the three *E. coli* strains, IC188, IC203 and IC5233. t-BOOH induced concentration-dependent cell death in all strains, with the wild-type strain IC188 being the least sensitive (IC50 = 201.37 mM). The sensitivity to t-BOOH was similar in the oxyR mutant IC203 (IC50 = 199.67 mM) when compared to the triple mutant IC5233 (IC50 = 172.98 mM).

The data of the IC50 curve show that t-BOOH concentrations lower than 25 mM were sublethal to the bacterial strains. Accordingly, we established the t-BOOH concentration range 0–25 mM to assess the oxidative stress induced by this compound. As for the experiments with H_2_O_2_, the analysis of intracellular ROS was restricted to living cells. 

The quantification of intracellular ROS in the wild-type strain IC188 and the strains deficient in *oxyR* (strain IC203) or in *oxyR*, *sodA* and *sodB* (strain IC5233) exposed to exogenous t-BOOH is shown in Figure 7.

As seen in Figure 7, the peroxidative activity induced by t-BOOH, a water-soluble alkyl hydroperoxide, was detected in all strains with both fluorescent probes, H_2_DCF-DA and MitoPY1, with both probes showing low fluorescence emission above autofluorescence in the absence of exogenous t-BOOH. In contrast to what we observed for H_2_O_2_, H_2_DCF-DA (Figure 7A) was much more sensitive to t-BOOH than MitoPY1 (Figure 7B). All the strains showed significant differences in the H_2_DCF-DA fluorescence ratio at all t-BOOH concentrations tested, while the MitoPY1 fluorescence ratio changed significantly only from 5 mM up to the highest t-BOOH concentration tested. H_2_DCF-DA was also better at revealing differences among the strains at all of the t-BOOH concentrations tested, with the triple mutant IC5233 strain being the most sensitive to exogenous t-BOOH, as expected.

Exposure to exogenous t-BOOH induced a dose-dependent increase in the HE fluorescence ratio in all strains, without any significant difference among wild-type and mutant strains (Figure 7C). The increase in HE fluorescence ratio was lower than that observed for H_2_DCF-DA fluorescence, but slightly higher than that observed for the MitoPY1 probe under the same experimental conditions.

These results suggest that H_2_DCF-DA may be the choice fluorescent probe for detecting intracellular peroxidative activity of exogenous peroxides in bacteria and that intracellular O_2_^−^ may be generated in *E. coli* cells exposed to exogenous organic hydroperoxides [26].

#### 2.4.3. Effect of CHP Treatment

Figure 8 shows the results of the viability assay to determine the cytotoxicity of the lipid-soluble organic hydroperoxide CHP on the three *E. coli* strains IC188, IC203 and IC5233. CHP was much more toxic than H_2_O_2_ or t-BOOH and induced concentration-dependent cell death in all strains with similar potency, with the wild-type strain IC188 (IC50 = 7.45 mM) and the triple mutant strain IC5233 (IC50 = 7.32 mM) being slightly less sensitive than the oxyR mutant strain IC203 (IC50 = 6.21 mM).

The data of the IC50 curve show that CHP concentrations lower than 1 mM were sublethal to the bacterial strains. Thus, we established the t-BOOH concentration range 0–1 mM to assess the oxidative stress induced by this compound. As for the experiments with H_2_O_2_ and t-BOOH, the analysis of intracellular ROS was restricted to living cells. 

The quantification of intracellular ROS in the wild-type strain IC188 and the strains deficient in *oxyR* (strain IC203) or in *oxyR, sodA* and *sodB* (strain IC5233) exposed to exogenous t-BOOH is shown in Figure 9.

As seen in Figure 9, the peroxidative activity induced by CHP, an organic lipid-soluble aromatic peroxide, was detected in all strains with both fluorescent probes, H_2_DCF-DA and MitoPY1, with both probes showing low fluorescence emission above autofluorescence in the absence of exogenous CHP. Consistent with what we observed for t-BOOH (Figure 7), another hydroperoxide compound, H_2_DCF-DA (Figure 9A), was much more sensitive to CHP than MitoPY1 (Figure 9B). As seen, all the strains showed significant differences in the H_2_DCF-DA fluorescence ratio at both CHP concentrations tested, while the MitoPY1 fluorescence ratio changed very slightly under the same experimental conditions. 

The H_2_DCF-DA fluorescence ratio revealed significant differences between the triple mutant strain (IC5233) and the wild-type (IC188) and *oxyR*-deficient (IC203) strains at all the concentrations tested, but only could distinguish IC1523 and IC203 strains at the highest CHP concentration. MitoPY1 showed a similar trend, but at a much lower fluorescence ratios, and was unable to show differences in strain-specific responses to exogenous CHP.

As shown in Figure 9C, exposure to exogenous CHP induced an important, dose-independent increase in the HE fluorescence ratio in all strains, but without significant differences among wild-type and mutant strains. The increase in the HE fluorescence ratio was slightly higher than that observed for H_2_DCF-DA fluorescence, but much higher than the one seen for the MitoPY1 probe under the same experimental conditions.

These results confirm that H_2_DCF-DA may be the choice fluorescent probe for detecting intracellular peroxidative activity of exogenous peroxides in bacteria and that intracellular O_2_^−^ may be generated in *E. coli* cells exposed to exogenous organic hydroperoxides [26].

## 3. Discussion

The aim of the present study was to assess issues of specificity in fluorescent probes and the involvement of different ROS in a relevant model of oxidative stress, based on the action of several exogenous peroxides differing in water or lipid solubility and reactivity. Because of their relevance in oxidative stress studies, we chose to apply the widely used H_2_DCF-DA and HE and the relatively new MitoPY1 as ROS-sensitive fluorescent probes in different strains of *E. coli* WP2 deficient in key genes for antioxidant defense, namely *oxyR*, *sodA* and *sodB* [11,12,18,19,20]. Appendix A shows a scheme of the genetic modifications induced on wildtype strain IC188 to generate the strains IC203 (deficient in oxyR) and IC5233 (deficient in oxyR, sodA and sodB).

The very first point to address in functional cytometry is whether to avoid or quantify the influence of the probes on the experimental system. In the case of oxidative stress, all reduced fluorogenic substrates are subject to auto-oxidation, which usually produces singlet oxygen, superoxide, and by its dismutation, H_2_O_2_. If the auto-oxidation rate is significant, it may result in the artifactual detection of ROS and higher background, a problem especially important for probes such as HE [16]. The concentration of the probe is also relevant, as it may affect the stoichiometry of the process under study. In fact, the probes themselves may affect the activity of ROS-producing enzymes [16]. Finally, fluorescent probes at a high concentration may be toxic to the cells [16].

To minimize artifacts derived from an excessive probe concentration, we initially titrated all of the fluorescent reagents used in the study (Figure 2 and Figure 3). Fluorochrome titration is an essential procedure when setting up a florescence-based determination in flow cytometry in order to define their optimal concentration for staining [23]. By defining on bacteria the minimal concentration of a given fluorescent probe required for the sensitive detection of a given ROS, we could minimize the non-specific detection of ROS, as demonstrated by the low level of intracellular fluorescence observed in cells not exposed to exogenous peroxides (Figure 5, Figure 7 and Figure 9).

An important point in the study was to determine both the sensitivity and the specificity of the fluorescent probes, as it is becoming evident that probes once considered as specific for a given ROS may indeed react with a variety of ROS and free radical species [15,16]. For this purpose, we designed a rational experimental setup made of:

Three *E. coli* B strains with different capacity for H_2_O_2_ and O_2_^−^ detoxification.Three exogenous peroxides differing in solubility and reactivity.Three fluorescent reagents differing in optical properties and specificity for ROS.

A major drawback for flow cytometric studies of bacterial function is the structural barrier of the cell wall that limits the uptake of vital dyes and requires transient or permanent permeabilization [27]. These manipulations are time consuming and may affect the physiology of the bacterial cell and result in cell aggregation or lysis. Our group developed a series of genetically modified strains of *E. coli* B [18,19,20]. As confirmed by multi-omic analysis [21], *E.coli* B strains express constitutively an altered cell-wall lipopolysaccharide resulting in increased membrane permeability, improving the uptake of exogenous chemicals. We have previously shown that flow cytometric analysis of *E. coli* B strains is a convenient alternative for cytometric assays of bacterial function [11,12].

Oxidative stress responses coordinated by specific regulators ensure bacterial survival during exposure to ROS, either exogenous or generated during normal respiration. OxyR protects *E. coli* against normally lethal concentrations of hydrogen peroxide or against thermal killing [10]. The *oxyR* deficiency blocks the synthesis of antioxidant enzymes induced by oxidative stress and results in increased intracellular content of ROS [10,18,19,20]. 

Strain IC203, deficient in *oxyR*, and its *oxyR* proficient parent WP2 uvrA/pKM101 (strain IC188) are the basis for a bacterial reversion assay developed by our group, the WP2 Mutoxitest, applied successfully to evaluate the oxidative mutagenicity of a large series of chemical compounds [20]. We found that many oxidative mutagens could be recognized by their greater mutagenic response in IC203 than in IC188, including the three peroxides used in our study. Interestingly for our results, this previous study by our group demonstrated that mutagenesis by t-BOOH and CHP was not inhibited by catalase, indicating that secondary H_2_O_2_ generation was not involved in the oxidative mechanisms of these organic hydroperoxides [20]. These results are consistent with the higher sensitivity and specificity of the MitoPY1 probe for the detection of H_2_O_2_, as supported by its very low capacity for the organic hydroperoxides, as shown in Figure 5, Figure 7 and Figure 9. Indeed, MitoPY1 is a chemoselective fluorescent indicator of the arylboronate family, with improved selectivity for H_2_O_2_ over other ROS based on the selective H_2_O_2_-mediated transformation of arylboronates to phenols [28,29]. Thus, our overall data obtained with the IC203 strain allow us to extend the specificity of MitoPY1 to H_2_O_2_ already reported with mammalian cells [30] to a controlled bacterial model of intracellular peroxidative activity.

On the other hand, the results presented in Figure 5, Figure 7 and Figure 9 confirm the caveats regarding H_2_DCF-DA, a fluorescent probe most widely used for detecting intracellular oxidative stress [16]. Intracellular H_2_DCF is assumed traditionally to be oxidized by H_2_O_2_ and organic peroxides and has been used for assaying peroxides. However, H_2_DCF does not react directly with H_2_O_2_ in the absence of peroxidases, and the intracellular fluorescence of its product DCF is not a direct measure of H_2_O_2_ [15,31]. Even if H_2_DCF oxidation also occurs by means of the action of H_2_O_2_ or O_2_^−^ in the presence of Fe^2+^, the strong OH^.^ radical species is responsible for such oxidation. This is in accordance with the fact that in the IC203 strain, catalase-sensitive oxidative mutagens were poor inducers of mutations derived from 8-oxoguanine lesions, whereas organic hydroperoxides efficiently induced such mutations [20].

Our data (Figure 5, Figure 7 and Figure 9) show that H_2_DCF-DA is more sensitive for the organic peroxides t-BOOH and CHP than for H_2_O_2_, confirming the lack of selectivity of H_2_DCF for H_2_O_2_. Therefore, this probe should be used rather than MitoPY1 for studies involving organic peroxides acting through H_2_O_2_-indepent mechanisms.

Including strain IC5233 (WP2*uvrA* Δ*oxyR*/*sodAB*-/pKM101) in our study increased the relevance of our data. Isoenzymes of SOD are key antioxidant enzymes that are associated with different metal co-factors. The deletion of the *sodA* and *sodB* genes in addition to oxyR makes strain IC5233 hypersensitive with respect to mutability by H_2_O_2_ and superoxide, because of the intracellular accumulation of both H_2_O_2_ and O_2_^−^ following treatment with a wide range of prooxidants. Accumulation of both prooxidants may lead to the generation of the strong oxidant OH^.^ radical through the Haber–Weiss reaction, catalyzed by Fe3+ ions [15,16]. Consistent with this mechanism, the strain IC5233 produced the highest changes in fluorescence ratios of both H_2_DCF-DA and MitoPY1 probes (Figure 5, Figure 7 and Figure 9), for all exogenous peroxides, except for H_2_O_2_ detection by H_2_DCF-DA. 

These results further support the utility of the series of mutant strains of *E. coli* B developed in our laboratory and allow us to suggest two alternative combinations of strain/fluorescent probe suitable for in vitro studies of peroxidative activity by flow cytometry. On the one hand, H_2_O_2_-dependent peroxidative processes may be investigated using the IC5233 strain and the fluorescent probe MitoPY1. On the other hand, H_2_O_2_-independent mechanisms may be investigated with the IC5233 strain and H2DCF-DA.

The analysis of the results obtained with the O_2_^−^-sensitive probe HE (Figure 5, Figure 7 and Figure 9) provides interesting insights into both the specificity of this probe and the involvement of O_2_^−^ radicals in the actions of the exogenous peroxides tested. HE is the most popular fluorogenic probe used for detecting intracellular O_2_^−^ radical [16]. The reaction between O_2_^−^ and HE generates a highly specific red fluorescent product, 2-hydroxyethidium (2-OH-E+). However, in biological systems, another red fluorescent product, ethidium (E+), is also formed, usually at a much higher concentration than 2-OH-E+ [16].

While most previous reports indicate that HE does not react readily with H_2_O_2_ in cell-free solution, organic peroxides are able to oxidize HE with the formation of a fluorescent product in the presence of complexes of iron and/or heme proteins (e.g., cytochromes) or in cellular systems [16]. In addition, HE can be oxidized by H_2_O_2_ in the presence of iron or copper via the OH^.^ radical or higher oxidants of iron. Such specific formation of 2-OH-E+ during the oxidation of HE by Fenton’s reaction is SOD-inhibitable, and so the formation of 2-OH-E+ in this system should be attributed to O_2_^−^ formation [16]. This assumption is not consistent with the lack of significant effect of the OxyR and SodA/SodB deficiencies in the fluorescence ratio of HE after exposure to t-BOOH or CHP (Figure 7 and Figure 9). As seen, the IC5233 triple mutant, in which the levels of both H_2_O_2_ and O_2_^−^ are the highest, shows similar or lower HE fluorescence ratios than the oxyR deficient strain IC203 and, more surprisingly, the wild-type strain IC188 after treatment with organic peroxides. In both cases, however, the fluorescence ratios are dependent on the concentration of exogenous peroxides, pointing towards a O_2_^−^-independent oxidation of HE to form E+, as the fluorescent species. Only after treatment with H_2_O_2_ could we observe both a dose-dependent and a strain-dependent increase in the HE fluorescence ratio, suggesting that in these conditions, superoxide-dependent oxidation of HE was involved and the specific generation of 2-OH-E+ might explain the slight difference between the IC5233 and IC203 response (Figure 5). See Appendix A, for a scheme of the oxidative processes we believe are involved in the interaction among the fluorescent probes and the peroxides used in this study.

In order to understand the effects of active molecules with such different physical-chemical characteristics as H_2_O_2_ and the organic peroxides t-BOOH and CHP, it is important to consider the role of their lipophilicity and their ability to penetrate or transport through membranes. In this regard, a previous study [32] demonstrated kinetic differences in the oxidative effects of CHP and H_2_O_2_ in a simple erythrocyte model. At comparable oxidant concentrations, the oxidative stress induced in the membrane was always much higher for CHP, while intracellular oxidative effects were observed with H_2_O_2_. The intracellular action of CHP was gradual, generating radicals that produced membrane-initiated oxidative stress, which increased over time to a maximum and then decreased. On the contrary, H_2_O_2_ reacted rapidly, generating radicals highly reactive to intracellular components as main targets. H_2_O_2_-induced oxidative stress was maximal immediately after addition and rapidly disappeared.

The relative lipophilicity of a chemical compound is expressed by its partition coefficient (Kow), which is the concentration ratio of the compound between the aqueous phase (w) and the organic phase (o) in a two-solvent system (typically, octanol and water), immiscible at equilibrium.

Table 2 allows for comparing the lipophilicity of the peroxides used in our study and their cytotoxicity on *E.coli* strains. The data show a clear positive correlation between the lipophilicity of the peroxides and their toxicity to *E. coli*, suggesting greater quantitative importance of the peroxidative effects on the bacterial membrane and/or greater efficiency of the protection systems (*oxyR*-dependent or independent) against the intracellular effects of H_2_O_2_ than against the membrane oxidative stress induced by organic peroxides.

The results of our study aim to systematically assess issues of specificity in fluorescent probes and the involvement of ROS in a bacterial model of oxidative stress, extending the previous findings of McBee et al. [13]. Their data showed that properly controlled flow cytometry using selective fluorescent probes resulted in precise and accurate analysis of ROS generation and metabolic changes in stressed bacteria. In addition, our results may be relevant to prevent or minimize possible sources of error, and to provide recommendations for the proper design of cytometric studies of oxidative stress, in accordance with current recommendations and guidelines [16,23].

## 4. Materials and Methods

### 4.1. Bacterial Strains and Culture Conditions

The genetically modified strains of *E. coli* B used in this study were the strains IC188 (WP2*uvr*A *oxyR*^+^/pKM101), IC203 (WP2*uvr*A Δ*oxyR*30/pKM101) and IC5233 (WP2*uvr*A Δ*oxyR*30 *sodAB*-/pKM101). All of these strains were obtained from the collection of the former Instituto de Investigaciones Citológicas of Valencia, now hosted in the Príncipe Felipe Research Center (Valencia, Spain) [18,19]. Overnight cultures were prepared from 100 μL of frozen permanent cultures inoculated into 9 mL of LB medium: NaCl, 10 g/L; Gibco Bacto tryptone (Thermo Fisher, Waltham, MA, USA), 10 g/L; Gibco Yeast extract (Thermo Fisher, Waltham, MA, USA), 50 g/L and incubated for 24 h at 37 °C. PROVIDERS OF MEDIA

### 4.2. Verification of Bacterial Phenotypes

The phenotype determined by the outer membrane of the different *E. coli* strains was verified using the rough specific bacteriophage C21. One hundred microliters of the overnight cultures was mixed with 3 mL of molten top agar: Bacto Agar (Becton Dickinson, Franklin Lakes, NJ, USA), 6 g/L; NaCl, 5 g/L) and poured into LA medium (20 g agar per liter of LB medium) in 10 cm Petri dishes. A drop of phage C21 suspension (obtained from Dr. P. Quillardet, Institut Pasteur, Paris, France) and a sterile 5 mm Whatman paper disk with 10 µL of 1 mg/mL Crystal Violet stain (CV, Sigma Aldrich, Saint Louis, MO, USA) were placed on separate areas of the lawn of cells and the plates were incubated overnight at 37 °C. *E. coli* K-12 was resistant to the C21 phage and the CV stain, and *E. coli* B wild-type was sensitive to the C21 phage and resistant to the CV stain [18,19,20].

### 4.3. Verification of Strain Sensitivity to Peroxidative Xenobiotics 

The sensitivity of control strain IC188 and deficient strains IC203 and IC5233 was evaluated by disk determination of plate growth inhibition, as follows. One hundred microliters of a stationary culture was mixed with 3 mL of soft agar and the different bacterial strains seeded on plates with LA medium. Sterile Whatman paper disks (6 mm diameter) were placed on the surface of the plates impregnated with 300 μg/disk of each compound: H_2_O_2_, t-BOOH and CHP, obtained from Sigma Aldrich (Saint Louis, MO, USA). The growth inhibition halo around the filter was measured after the incubation of plates for 24 h at 37 °C [18,19].

### 4.4. Treatment with Peroxidative Xenobiotics for Flow Cytometry Analysis

Bacteria were grown in LB medium until the culture reached the mid-exponential growth phase, with an optical density at 600 nm (OD) of approximately 0.6. Bacterial cultures in the exponential state were diluted at 1/10 with LB culture medium and treated for 60 min at 37 °C with concentrations of H_2_O_2_ (0–100 mM), t-BOOH (0–25 mM) or CHP (0–1 mM). After treatment, cultures were washed once, resuspended in LB with the appropriate fluorescent probes and analyzed by means of flow cytometry [11,12].

### 4.5. Fluorescent Probes and Procedures for Assesing Bacterial Viability by Flow Cytometry

Determining cell viability is an important step when evaluating a cell’s response to prooxidants. Dead cells can generate artifacts because of non-specific staining with probes or, conversely, by impaired intracellular retention of fluorescent products [16]. The viability fluorescent probes propidium iodide (PI, Sigma Aldrich, Saint Louis, MO, USA; cat. P4170) and Helix NP^TM^Green (BioLegend, San Diego, CA, USA; cat. 425303) were stored frozen as stock solutions, as follows: Helix NP^TM^ Green (5 mM in DMSO) and PI (1.50 mM in dH2O). The final assay concentration was 0.50 µM Helix NP^TM^ Green and 7.5 µM PI.

PI and Helix NP^TM^Green are membrane-impermeant dyes excluded from viable cells. Both bind to double stranded DNA by intercalating between base pairs. PI is excited at 488 nm and emits at a maximum wavelength of 617 nm [35]. Helix NP^TM^Green is a green-emitting dye with excitation/emission at 495 nm/519 nm [36].

Bacterial cultures in an exponential state were diluted at 1/10 with LB medium and treated for 60 min at 37 °C with H_2_O_2_ (0 to 1200 mM), t -BOOH (0 to 200 mM) and CHP (0 to 10 mM). The cultures treated with the prooxidants were centrifuged for 10 min at 4500× *g* and resuspended in 500 µL LB. PI or Helix NP^TM^ Green (for HE-stained samples) were added to sample tubes and incubated for 5 min immediately prior to flow cytometric analysis.

### 4.6. Fluorescent Probes and Staining Procedures for ROS Detection by Flow Cytometry

The following fluorescent probes were purchased from Sigma Aldrich (Saint Louis, MO, USA): 2′,7′-dichlorodihydrofluorescein diacetate, also known as 2′,7′-dichlorofluorescin diacetate (H_2_DCF-DA, cat. D6883); dihydroethidium, also known as hydroethidine (HE, cat. D7008); mitochondrial peroxy yellow 1 (MitoPY1, cat. SML0734); and propidium iodide (PI, cat. P4170). The fluorescent probe Helix NP^TM^Green was purchased from BioLegend (San Diego, CA, USA; cat. 425303). The fluorescent probes were stored frozen as stock solutions at the indicated concentrations in DMSO: H_2_DCF-DA (20.5 mM); HE (3.17 mM); MitoPY1 (1.05 mM) and Helix NP Green (5 mM). PI was stored frozen at 1.50 mM in dH_2_O. The final assay concentration for each fluorochrome was as follows: H_2_DCF-DA: 41.04 µM; HE: 15.88 µM; MitoPY1: 5 µM; Helix NP Green: 0.50 µM; and PI: 7.5 µM.

The cell-permeant H_2_DCF-DA is one of the most commonly fluorogenic substrates used for studies related to ROS generation. Upon cleavage of its acetate groups by intracellular esterases, oxidation of intracellular 2′,7′- dichlorodihydrofluorescein (DCFH) produces 2′,7′ dichlorofluorescein (DCF), a fluorescent product with excitation at 498 nm and emission at 522 nm [37,38].

MitoPY1 is a boronate-based fluorescent probe that selectively targets mitochondria and senses local H_2_O_2_ generation [39]. MitoPY1 can be excited by the 488 argon-ion laser, and its green fluorescence is detected at around 520 nm [29].

HE has been widely used as a fluorogenic substrate to detect O_2_**^−^** [16]. HE is membrane-permeable and once oxidized by O_2_^−^, it generates a fluorescent compound retained in the nucleus by intercalating with DNA, which increases its fluorescence, with excitation at 520 nm and orange emission at 610 nm [24].

Bacterial cultures in an exponential state were diluted at 1/10 with LB medium and treated for 60 min at 37 °C with H_2_O_2_ (0 to 100 mM), t -BOOH (0 to 25 mM) and CHP (0 to 1 mM). The cultures treated with the prooxidants were centrifuged for 10 min at 4500× *g* and resuspended in 500 µL LB medium. The fluorescent probes H_2_DCF-DA, MitoPY1 or HE were added, and samples were incubated for 30 min at 37 °C in the dark [11,12]. PI or Helix NP^TM^ Green (for HE-stained samples) were added for 5 min prior to flow cytometric analysis.

### 4.7. Titration of ROS-Sensitive Fluorescent Probes for Flow Cytometry Analysis

When setting up a fluorescence-based determination in flow cytometry, it is essential to titrate the fluorescent probes properly in order to define their appropriate concentration for staining, thus minimizing saturation and suboptimal detection of the relevant biological parameters [23]. To define the final concentration of the different fluorochromes, we selected the most appropriate oxidant, according to the preferential sensitivity of the fluorescent probe. Thus, for the probes H_2_DCF-DA and MitoPY1, cell suspensions were exposed to t-BOOH. For the titration of the probe HE, cell suspensions were treated with menadione, which generates O_2_^−^ by redox cycling [22]. 

Bacterial cultures in an exponential state were diluted at 1/10 with LB medium and treated for 60 min at 37 °C with 10 mM t-BOOH or for 120 min with 0.29 mM menadione. The cultures were then centrifuged for 10 min at 4500× *g* and resuspended in 500 µL LB medium. The fluorescent probes H_2_DCF-DA (0 to 164.16 µM), MitoPY1 (0 to 10 µM) or HE (0 to 23.81 µM) were added and samples were incubated for 30 min at 37 °C in the dark. PI or HelixNP Green (for HE-stained samples) were added 5 min immediately prior to flow cytometric analysis.

### 4.8. Flow Cytometric Analysis

As a general principle, we followed the current guidelines and recommendations for performing flow cytometric analysis [23].

Parameter selection and cytometer settings were optimized for analysis on a Gallios flow cytometer (Beckman Coulter, Brea, CA, USA) equipped with three lasers emitting at 405, 488 and 638 nm. The fluorescent intracellular products of the probes H_2_DCF-DA, MitoPY1 and Helix NP^TM^ Green were excited by the blue laser (488 nm) and detected with a 525/40 nm band-pass emission filter. HE and PI were detected with a 620/30 nm band-pass emission filter after excitation by the blue laser.

Prior to analysis, unlabeled bacteria were run to optimize voltage settings, and the flow rate was set on low. Bacterial events were discriminated from debris using forward scatter (FSC) and side scatter (SSC) signals. Doublets were excluded from analysis by FSC-A and FSC-peak. Gates applied for population discrimination were set manually based on control samples. For each sample, 20,000 ungated events were collected. For further analysis, the cytometric data were exported to Kaluza 2.1 software (Beckman Coulter, Brea, CA, USA). 

The mean fluorescence intensities (MFI) were expressed as relative fluorescence units (RFU). To obtain normalized fluorescence ratio values, the fluorescence intensity of the treated samples was divided by the fluorescence intensity of the basal condition. The results of the different experiments involving fluorescence quantitation shown are the averages ± SD of the ratios for three separate experiments. 

In viability experiments, the cytotoxic potency of xenobiotics was determined by their IC50 values, i.e., the concentration of xenobiotic at which cell viability is inhibited by 50%, as calculated by curve fitting and determination by the correlation coefficient (R2).

### 4.9. Statistical Analysis

Statistical analysis was carried out with GraphPad Prism (La Jolla, CA, USA) version 9.0 software for Windows and flow cytometric graphs were generated using Kaluza analysis software.

In order to choose the appropriate statistical test to carry out the hypothesis contrast, we previously checked the normality in the groups. Normality was determined using the Shapiro–Wilk test for sample size. When the assumption of normality was met, we checked whether there was homoscedasticity or equality of variances using the Levene test. With equal variances, the t-Student test for two independent samples was applied in the comparisons between the control strain IC188 and the mutant strains. For comparisons between more than two groups, the one-way analysis of variance (ANOVA) test was performed, with Dunnett’s multiple comparisons. This test was applied to study the differences in the concentrations of each fluorochrome with respect to its corresponding baseline. Statistical significance was considered when *p* < 0.05. 

## 5. Conclusions

This is a systematical flow cytometry study aiming to assess issues of specificity in fluorescent probes and the involvement of different ROS in a model of oxidative stress on genetically engineered *E. coli* strains. Our results may be relevant to prevent or minimize possible sources of error, and to provide recommendations for the proper design of cytometric studies of oxidative stress, in accordance with previous studies [11,12,13] and current recommendations and guidelines [16,23].

## Figures and Tables

**Figure 1 ijms-23-06537-f001:**
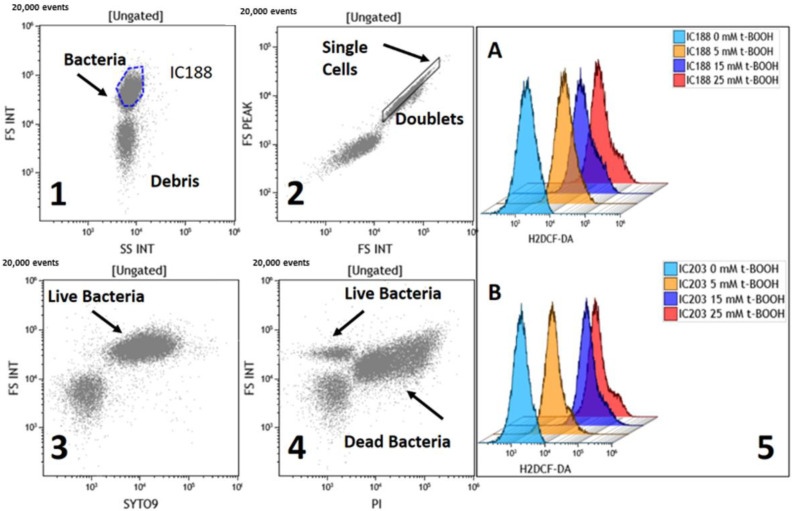
Scheme of the flow cytometric criteria to identify and gate single live bacteria for the analysis of the intracellular level of ROS. (**1**) shows the light-scatter features (FS Int vs. SS Int) allowing us to define the gate for bacteria, distinguished from debris. (**2**) shows the gate created to include only single cells in the further analysis. (**3**,**4**) confirm that the light-scatter features of the events in the gate “bacteria” in (**1**) identify live bacteria as events able to be stained with SYTO-9 (**3**); data from an untreated sample) and to become labelled with PI in conditions of cytotoxicity (**4**); data from cells treated with 10 mM t-BOOH). (**4**) also shows that the fraction of live cells in toxic-treated samples can be identified and gated for analyzing intracellular fluorescence associated with ROS generation, as shown in (**5**) for two samples of wild-type (IC188, (**A**)) and *oxyR*-deficient (IC203, (**B**)) *E. coli* strains exposed to different concentrations of t-BOOH.

**Figure 2 ijms-23-06537-f002:**
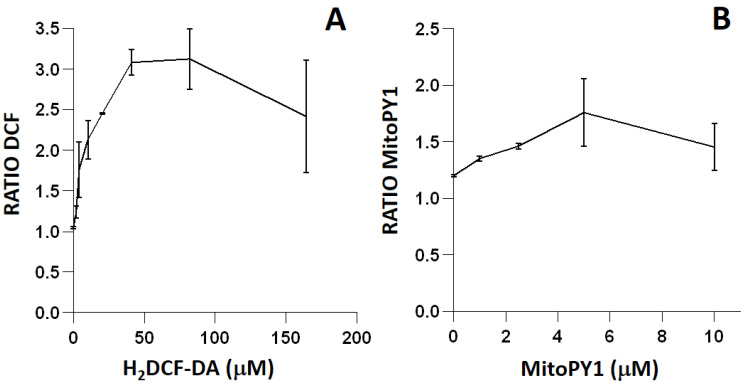
Titration of the fluorescent probes H_2_DCF-DA (**A**) and MitoPY1 (**B**) in strain IC188 exposed to 10 mM t-BOOH. Results are the mean ± SD (n = 3) of the fluorescence ratios of the products resulting from intracellular probe oxidation.

**Figure 3 ijms-23-06537-f003:**
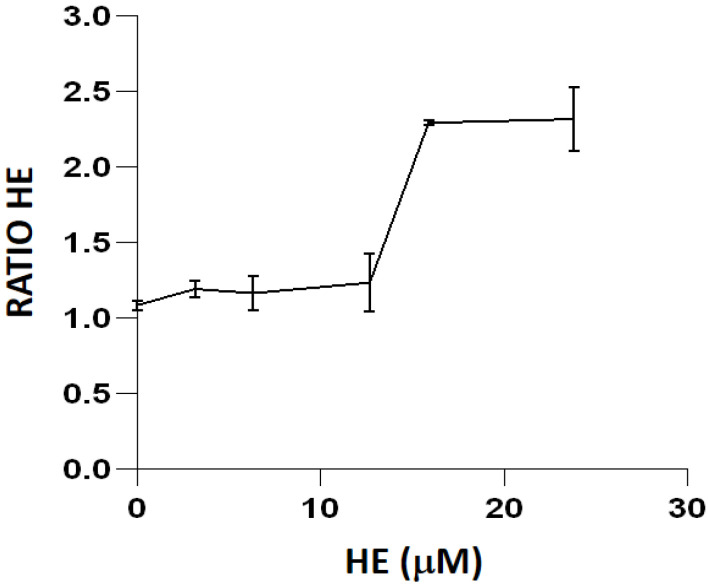
Titration of the fluorescent probe HE in strain IC5233 exposed to 0.29 mM menadione. Results are the mean ± SD (n = 3) of the fluorescence ratios of the product resulting from intracellular probe oxidation.

**Figure 4 ijms-23-06537-f004:**
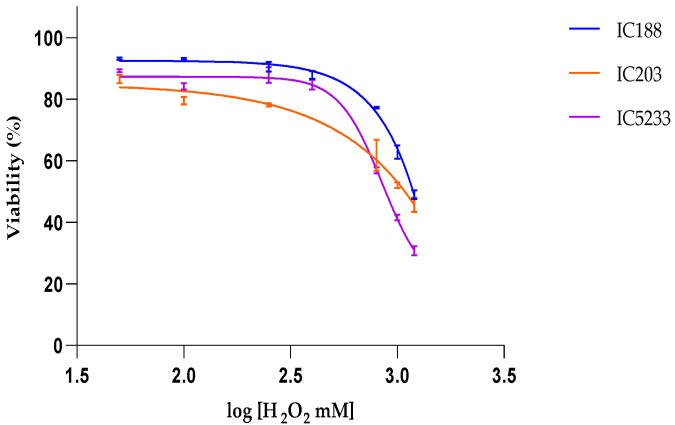
Flow cytometric assay of H_2_O_2_ cytotoxicity to *E. coli* strains IC188, IC203 and IC5233. The curves represent the % of live cells versus the logarithm of the concentration. Results are mean ± SD (n = 3).

**Figure 5 ijms-23-06537-f005:**
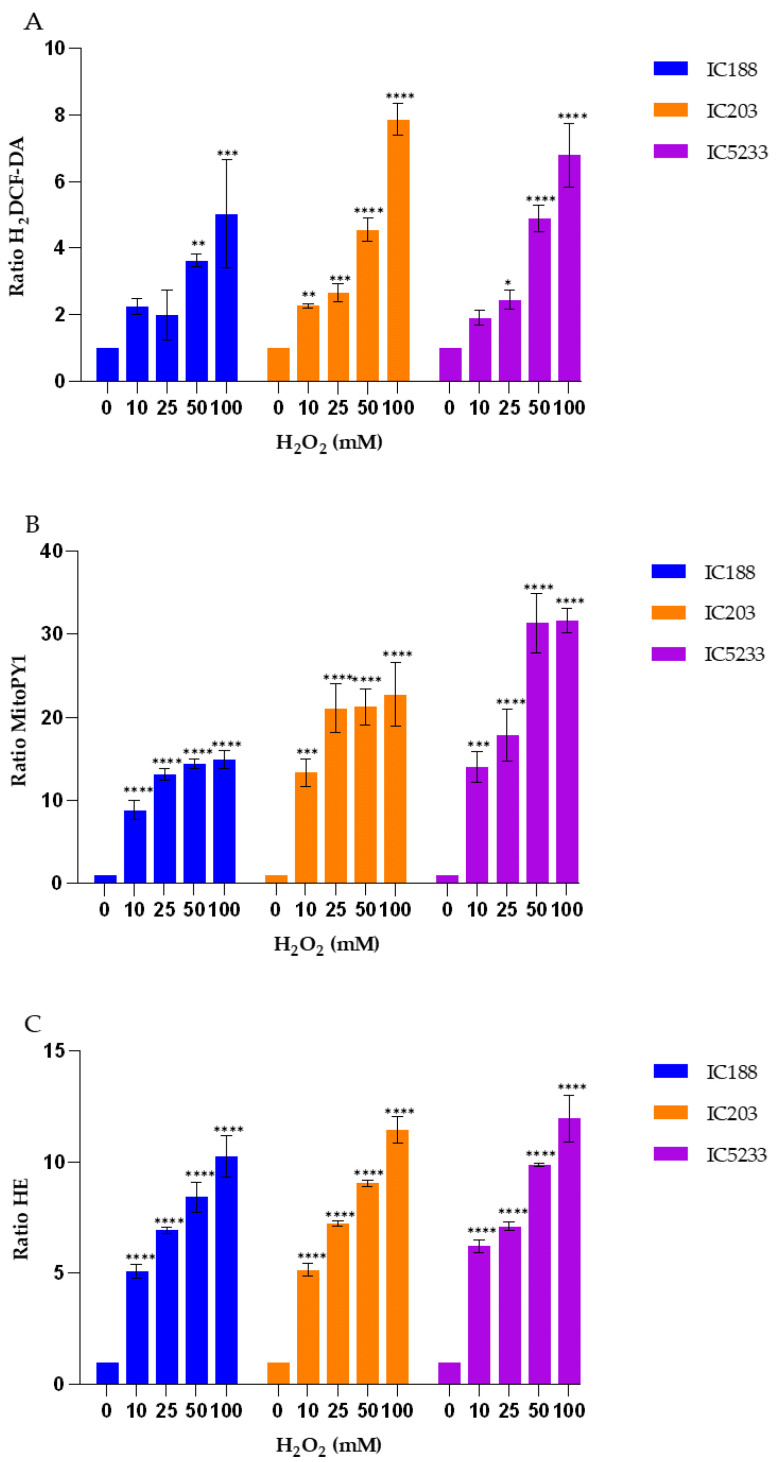
Fluorescence response to exogenous H_2_O_2_ of the fluorescent probes (**A**) H_2_DCF-DA, (**B**) MitoPY1 and (**C**) HE in the strains IC188, IC203 and IC5233. Results are the mean ± SD (n = 3) of the fluorescence ratios of the product resulting from intracellular probe oxidation. (* *p* < 0.05, ** *p* < 0.01, *** *p* < 0.001 and **** *p* < 0.0001 in Anova test).

**Figure 6 ijms-23-06537-f006:**
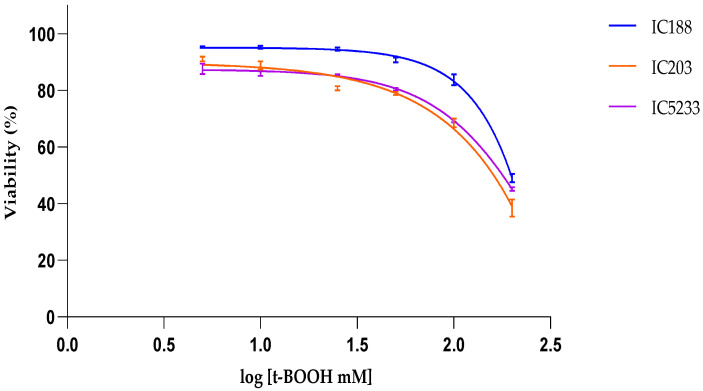
Flow cytometric assay of t-BOOH cytotoxicity to *E. coli* strains IC188, IC203 and IC5233. The curves represent the % of live cells versus the logarithm of the concentration. Results are mean ± SD (n = 3).

**Figure 7 ijms-23-06537-f007:**
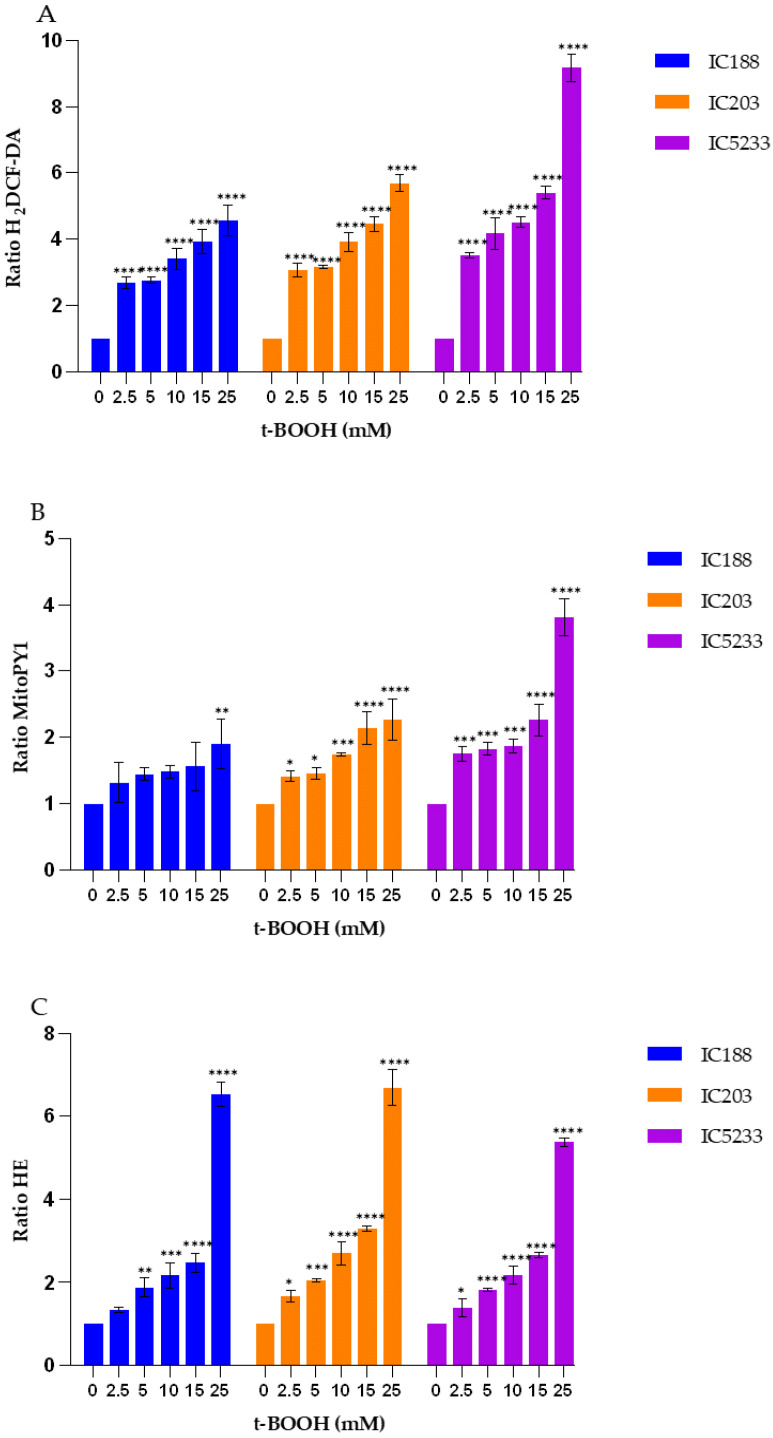
Fluorescence response to exogenous t-BOOH of the fluorescent probes (**A**) H_2_DCF-DA, (**B**) MitoPY1 and (**C**) HE in the strains IC188, IC203 and IC5233. Results are the mean ± SD (n = 3) of the fluorescence ratios of the product resulting from intracellular probe oxidation. (* *p* < 0.05, ** *p* < 0.01, *** *p* < 0.001 and **** *p* < 0.0001 in Anova test).

**Figure 8 ijms-23-06537-f008:**
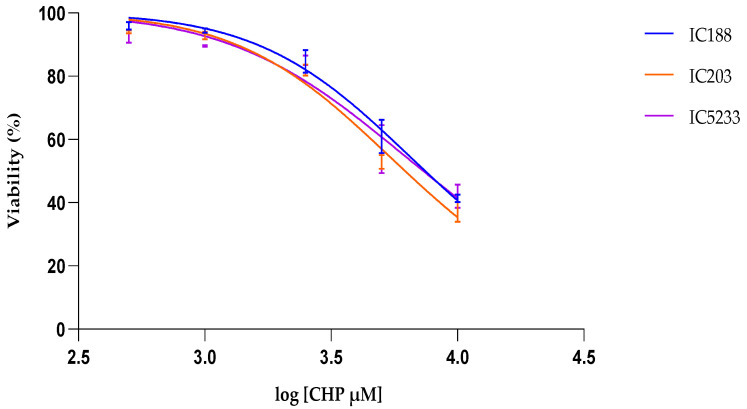
Flow cytometric assay of CHP cytotoxicity to *E. coli* strains IC188, IC203 and IC5233. The curves represent the % of live cells versus the logarithm of the concentration. Results are mean ± SD (n = 3).

**Figure 9 ijms-23-06537-f009:**
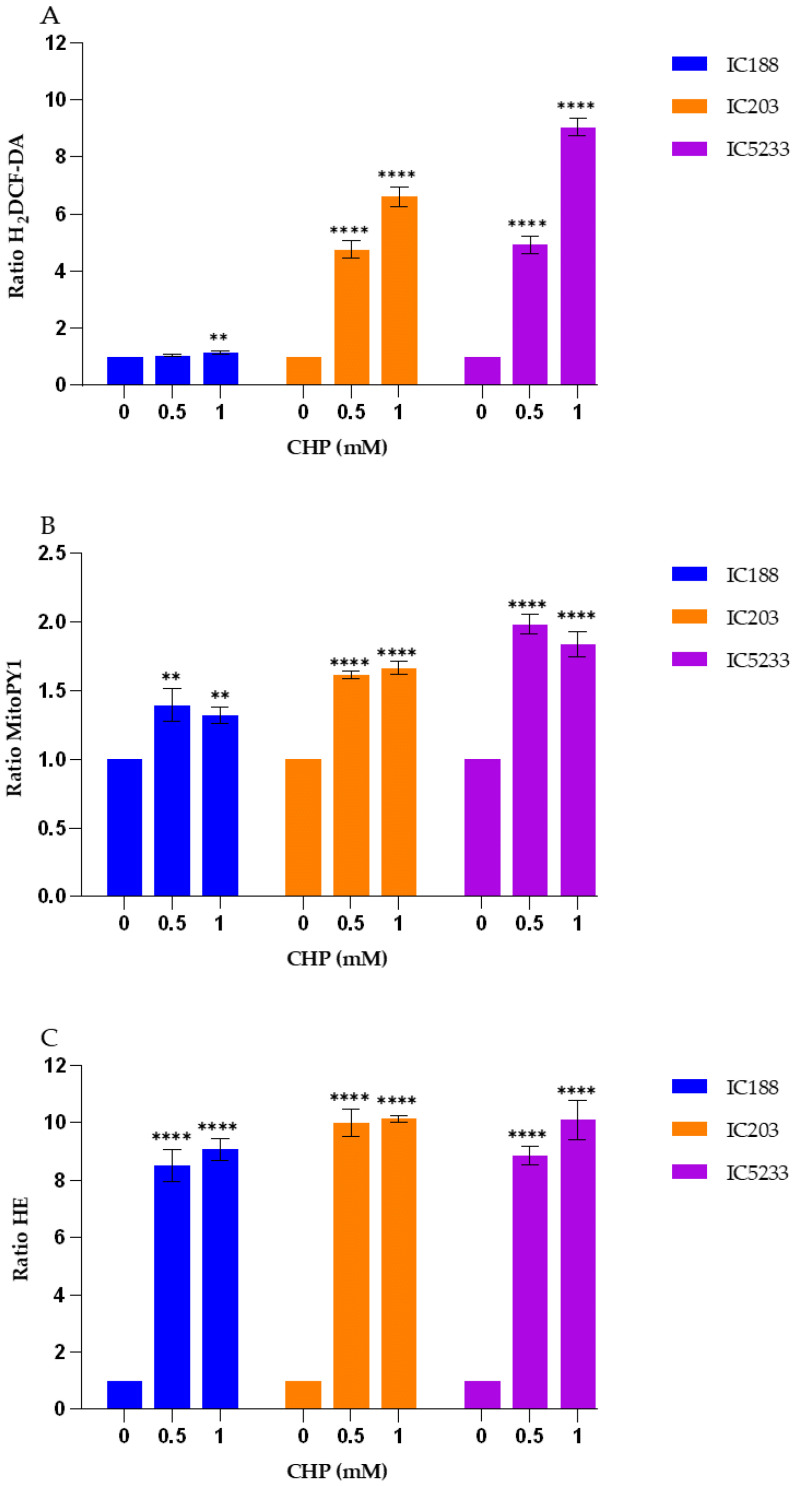
Fluorescence response to exogenous CHP of the fluorescent probes (**A**) H_2_DCF-DA, (**B**) MitoPY1 and (**C**) HE in the strains IC188, IC203 and IC5233. Results are the mean ± SD (n = 3) of the fluorescence ratios of the product resulting from intracellular probe oxidation. (** *p* < 0.01 and **** *p* < 0.0001 in Anova test).

**Table 1 ijms-23-06537-t001:** Sensitivity of the different strains to oxidants in the growth inhibition test.

Strains	Compounds			
	H_2_O_2_	t-BOOH	CHP	Menadione
IC188	1.77 ± 0.06	0.63 ± 0.06	1.97 ± 0.06	0.87 ± 0.06
IC203	2.87 ± 0.12	2.97 ± 0.25	2.67 ± 0.06	2.30 ± 0.10
IC5233	2.90 ± 0.49	0.67 ± 0.06	2.63 ± 0.15	2.67 ± 0.15

Bacterial lawns on Petri dishes were exposed for 24 h to the compounds (300 μg/disc). Results are mean ± SD (n = 3) of the growth inhibition halo in mm.

**Table 2 ijms-23-06537-t002:** Comparison between the lipophilicity (Kow) of the model compounds employed and their cytotoxicity in *E.coli* B strains.

Compound	Kow (Log P)	IC188 (IC_50_)	IC203 (IC_50_)	IC5233 (IC_50_)	Reference
H_2_O_2_	−1.36	1342.76	1330.45	847.23	[33]
t-BOOH	0.94	201.37	172.98	199.67	[34]
CHP	2.16	7.45	6.21	7.32	[30]

The partition coefficient values are expressed in logarithmic form (Log P) and IC50s in mM. The partition coefficient values were obtained from the PubChem Compound Summary repository [30,33,34]. IC50 values were derived from the data in Figure 4, Figure 6 and Figure 8.

## Data Availability

The data presented in this study are available on request from the corresponding author.

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
