# Peer review of "Flow Cytometric Analysis of Oxidative Stress in Escherichia coli B Strains Deficient in Genes of the Antioxidant Defence"

_ijms, 2022, doi:10.3390/ijms23126537_

Round 1

Reviewer 1 Report

The authors submitted quite fine paper focused on flow cytometry study aimed to assess systematically issues of specificity of fluorescent probes and involvement of different ROS in a  novel bacterial model of oxidative stress. In my opinion this paper needs minor improvement.

Minor concerns:

  1. Methodology is the study of research methods or more formally, "'a contextual framework' for research, so flow cytometry cannot be methodology of choice it is method of choice.
  2. “For each sample, 20000 events were collected.” Please specify if this was total number of events or in specific gate.
  3. In figures 5, 7 and 9 legends please describe results of what comparison are represented with * it makes the figures independent form text and make them more soundy for readers
  4. The resolution of figures 2 and 3 shall be improved, if this is final version and resolution wasn’t lost during conversion to pdf.
  5. Please check carefully “Materials and Methods” section, because in some case manufacturers name or city and country of manufacturer’s origin are missing

Author Response

RESPONSE TO REVIEWER 1

Comments and Suggestions for Authors:

The authors submitted quite fine paper focused on flow cytometry study aimed to assess systematically issues of specificity of fluorescent probes and involvement of different ROS in a novel bacterial model of oxidative stress. In my opinion this paper needs minor improvement.

Response: We thank very much the reviewer for his opinion and suggestions.

Minor concerns:

Methodology is the study of research methods or more formally, "'a contextual framework' for research, so flow cytometry cannot be methodology of choice it is method of choice.

Response: We have replaced “methodology” with “method” (line 55).

“For each sample, 20000 events were collected.” Please specify if this was total number of events or in specific gate.

Response: This has been clarified. We collected 20,000 ungated events (line 586).

In figures 5, 7 and 9 legends please describe results of what comparison are represented with * it makes the figures independent form text and make them more soundy for readers.

Response: The legends of Figs. 5, 7 and 9 has been modified to incorporate the statistical information required.

The resolution of figures 2 and 3 shall be improved, if this is final version and resolution wasn’t lost during conversion to pdf.

Response: High-resolution versions of Figs 2 and 3 are now included.

Please check carefully “Materials and Methods” section, because in some case manufacturers name or city and country of manufacturer’s origin are missing.

Response: This whole section has been revised, and the city and country of manufacturers’ origin have been included.  

Reviewer 2 Report

The submitted manuscript titled: “Flow Cytometric Analysis of Oxidative Stress in Escherichia Coli B strains Deficient in Genes of the Antioxidant Defence” by Jávega, B.; et al. is an innovative work where three Escherichia Coli strains (oxyR, sodA and sodB) are employed to evaluate fluorescent probes specificity towards reactive oxygen species (ROS) detection. The most relevant outcomes of this work may significantly aid to develop new strategies on the design of cytometric studies of oxidative stress involving different biological sources. The gathered findings may be relevant for the examined field. The results achieved are well-discussed during the main body of the reported manuscript. The scientific paper is well written. In my opinion the present manuscript is innovative and the methodological approached used matches with the scope of International Journal of Molecular Sciences. For the above described reasons, I recommend the publication in International Journal of Molecular Sciences once the following remarks will be fixed:

--------

RESULTS

Result section is well-structured and clearly explained. Nevertheless, authors should pay attention to following aspects:

  1. Many data appear with comas instead of points. For example, the entire data from Table 1 (line 98) should be exchanged (e.g. “1.77 ± 0.06” in place of “1,77 ± 0,06”). In addition, “IC50 = 1,342 mM” (lines 180-181) and “IC50 of IC203 (1,430 mM) (line 184) should be also modified following the same recommendation.
  2. The standard deviation (SD) error of some points from Figure 2, Figure 3, Figure 4, Figure 6 and Figure 8 is not properly visualized. Authors should slightly decrease the point size in order to fix this issue. In case not to be possible (because the SD error is almost negligible, please create a table with all these data and introduce it as Supplementary Information).
  • In addition, the Y-axis from Figure 4, Figure 6 and Figure 8 should be shortened in order to increase its resolution (e.g. data from Fig. 2A and Fig. 2B should range from 0.0 to 3.5 and from 0.0 to 2.5, respectively).
  1. Significant figures must be homogenized for all data. For example, “IC50 = 201 mM” (line 231), “IC203 (IC50 = 199 mM)” (lines 231-232), “(IC50 = 173 mM)” (line 232) and data from Table 2 do not contain any decimal in contrast to Table 1 where two decimals are shown.
  2. Even if it is optional, I may include a schematic representation of all oxidative reactions that take place with the three fluorescent dyes. This schematic representation may be placed as Supplementary information. This information will significantly aid to the potential readers to better understand the chemical reactivity of the fluorescent dyes used under oxidative conditions.

--------

DISCUSSION

Authors used Escherichia Coli strands as model system. Authors broadly suggest during the entire manuscript body the submitted work is the first time that ROS assessment with fluorescent probes in bacterial model is reported. I was wondering to know the opinion of the authors regarding the following work:

McBee, M.E.; Chionh, Y.H.; Sharaf, M.L.; Ho, P.; Cai, M.W.L.; Dedon, P.C. Production of Superoxide in Bacteria Is Stress- and Cell State-Dependent: A Gating-Optimized Flow Cytometry Method that Minimizes ROS Measurement Artifacts with Fluorescent Dyes. Front. Microbiol. 2017, 8, 459. https://doi.org/10.3389/fmicb.2017.00459.

This article provides the following information: “These results demonstrate that properly controlled flow cytometry coupled with fluorescent probes provides precise and accurate quantitative analysis of ROS generation and metabolic changes in stressed bacteria” (Abstract section) and “We developed a four-step gating strategy to minimize false positive signals and artifacts by accounting for DNA content, changes in cell morphology, dye uptake and retention, and target specific dye activation in bacterial cells. This approach is generally applicable to intracellular metabolic probes, as illustrated here with CellROX Green, CTC and CFDA-AM” (Discussion section).

What is the opinion of the authors regarding this work? In case that previous studies are already available on literature, authors should change the claim that the submitted work is the first study to address the ROS with fluorescent probes by using bacterial models.

--------

BIBLIOGRAPHY

The bibliography is not in the proper format of IJNS journal. Authors must take care of this aspect and deeply revise this section. The name of the Journal and the volume should be indicated in Italics. The year of the publication must be remarked in bold. Then, authors name list should be also checked.

Author Response

Comments and Suggestions for Authors

The submitted manuscript titled: “Flow Cytometric Analysis of Oxidative Stress in Escherichia Coli B strains Deficient in Genes of the Antioxidant Defence” by Jávega, B.; et al. is an innovative work where three Escherichia Coli strains (oxyR, sodA and sodB) are employed to evaluate fluorescent probes specificity towards reactive oxygen species (ROS) detection. The most relevant outcomes of this work may significantly aid to develop new strategies on the design of cytometric studies of oxidative stress involving different biological sources. The gathered findings may be relevant for the examined field. The results achieved are well-discussed during the main body of the reported manuscript. The scientific paper is well written. In my opinion the present manuscript is innovative and the methodological approached used matches with the scope of International Journal of Molecular Sciences. For the above described reasons, I recommend the publication in International Journal of Molecular Sciences once the following remarks will be fixed.

Response: We thank very much the reviewer for his opinion and suggestions.

RESULTS

Result section is well-structured and clearly explained. Nevertheless, authors should pay attention to following aspects:

Many data appear with comas instead of points. For example, the entire data from Table 1 (line 98) should be exchanged (e.g. “1.77 ± 0.06” in place of “1,77 ± 0,06”).

Response: Table 1 has been rewritten and decimal commas replaced by decimal points (Table 1)

In addition, “IC50 = 1,342 mM” (lines 180-181) and “IC50 of IC203 (1,430 mM) (line 184) should be also modified following the same recommendation.

Response: These numbers are correct. Indeed, the IC50 values C very high for these bacterial strains, indicating a relatively low cytotoxicity for H2O2 as compared with the two organic hydroperoxides used in our study.

The standard deviation (SD) error of some points from Figure 2, Figure 3, Figure 4, Figure 6 and Figure 8 is not properly visualized. Authors should slightly decrease the point size in order to fix this issue. In case not to be possible (because the SD error is almost negligible, please create a table with all these data and introduce it as Supplementary Information).

Response: Figs. 2, 3, 4, 6 and 8 have been redrawn according to the reviewer suggestion in order to make clearly visible the error bars.

In addition, the Y-axis from Figure 4, Figure 6 and Figure 8 should be shortened in order to increase its resolution (e.g. data from Fig. 2A and Fig. 2B should range from 0.0 to 3.5 and from 0.0 to 2.5, respectively).

Response: Figs. 4, 6 and 8 have been redrawn according to the reviewer suggestion in order to increase their resolution.

Significant figures must be homogenized for all data. For example, “IC50 = 201 mM” (line 231), “IC203 (IC50 = 199 mM)” (lines 231-232), “(IC50 = 173 mM)” (line 232) and data from Table 2 do not contain any decimal in contrast to Table 1 where two decimals are shown.

Response: IC50 values have been homogenized to the second decimal in lines 231-232 and Table 2.

Even if it is optional, I may include a schematic representation of all oxidative reactions that take place with the three fluorescent dyes. This schematic representation may be placed as Supplementary information. This information will significantly aid to the potential readers to better understand the chemical reactivity of the fluorescent dyes used under oxidative conditions.

Response: We thank the reviewer for this excellent suggestion. We have included Supplemental Fig. 1 with a scheme of the genetic modifications induced on wildtype strain IC188 to generate the strains IC203 (deficient in oxyR) and IC5233 (deficient in oxyR, sodA and sodB) and Supplemental Fig.2, with  a scheme of the oxidative processes we believe are involved in the interaction among the fluorescent probes and peroxides used in this study.

DISCUSSION

Authors used Escherichia Coli strands as model system. Authors broadly suggest during the entire manuscript body the submitted work is the first time that ROS assessment with fluorescent probes in bacterial model is reported. I was wondering to know the opinion of the authors regarding the following work: McBee, M.E.; Chionh, Y.H.; Sharaf, M.L.; Ho, P.; Cai, M.W.L.; Dedon, P.C. Production of Superoxide in Bacteria Is Stress- and Cell State-Dependent: A Gating-Optimized Flow Cytometry Method that Minimizes ROS Measurement Artifacts with Fluorescent Dyes. Front. Microbiol. 2017, 8, 459. https://doi.org/10.3389/fmicb.2017.00459. This article provides the following information: “These results demonstrate that properly controlled flow cytometry coupled with fluorescent probes provides precise and accurate quantitative analysis of ROS generation and metabolic changes in stressed bacteria” (Abstract section) and “We developed a four-step gating strategy to minimize false positive signals and artifacts by accounting for DNA content, changes in cell morphology, dye uptake and retention, and target specific dye activation in bacterial cells. This approach is generally applicable to intracellular metabolic probes, as illustrated here with CellROX Green, CTC and CFDA-AM” (Discussion section). What is the opinion of the authors regarding this work?

Response: We were aware of this interesting paper and we cited this work among other previously published papers reporting ROS analysis in bacteria (including earlier work by our own lab). Indeed, our claim that “To our knowledge, this is the first flow cytometry study aimed to assess systematically issues of specificity of fluorescent probes and involvement of different ROS in a novel bacterial model of oxidative stress” intended to point out that our design was different to the experimental design of McBee et al. paper, that is centered on the accurate detection of superoxide radical in bacteria. In their paper, these authors evaluate only a single ROS-sensitive probe (CellRox Green) and two ROS-generating systems (H2O2 and Menadione). The two other fluorescent probes were not directly related to oxidative stress (CFDA for esterase activity and CTC for cellular respiratory activity, respectively) and were used as proofs of concept of their gating strategy.    

In case that previous studies are already available on literature, authors should change the claim that the submitted work is the first study to address the ROS with fluorescent probes by using bacterial models.

Response: In spite of our comments above, and to avoid excesive affirmations, we have reduced the strength of our claim in the Abstract, Discussion and Conclusions sections and have included citation and a specific comment on McBee et al. paper in the Discussion section.

BIBLIOGRAPHY. The bibliography is not in the proper format of IJNS journal. Authors must take care of this aspect and deeply revise this section. The name of the Journal and the volume should be indicated in Italics. The year of the publication must be remarked in bold. Then, authors name list should be also checked.

Response: The Bibliography section has been carefully checked for adjusting to the journal format.

Reviewer 3 Report

In this work, the authors have measured Oxidative stress in E. coli using different fluorophore in B-type wild type strain and strain lacking the arsenal to mitigate oxidative stress namely oxyR alone or oxyR along with Superoxide dismutase sodA and sodB. The authors have titrated the amount to fluorophore and oxidizing agent in such a fashion that all the experiments were conducted in the sublethal range of the oxidizing agents. Using the flow cytometric assay, the authors were able to show dose dependent response to oxidizing agents. Mito-PY1 showed the greatest sensitivity when the bacterial strains were challenged with H2O2, whereas DHDCF and HE outperformed other dyes in response to t-BOOH and CHP. Although, the authors were able to measure reactive oxygen species using the fluorescence dye, the study failed to replicate the well-established fact in the field of oxidative stress response: deletion of sodA and sodB leads to oxidative stress response. In most cases, the authors showed there was a minor difference between in sensitivity between wild type strain and strain lacking oxyR, soda and sodB. For example, when challenged with H2O2, there was less than 2-fold change in the fluorescence intensity of Mito-PY1 between wild type and the strains lacking the superoxide dismutase. This was rather bizarre as treating the cells with H2O2 should have caused significant oxidative stress in strains lacking soda and sodB.

The authors should clarify the following points:

  1. Why was less oxidative stress detected in a strain lacking oxyR, sodA and sodB. It is well known in the filed of oxidative stress response that deletion of sodA and sodB leads severe growth defect due to oxidative stress that is more than 1000-fold compared to wild type. Whereas the authors have failed to any significant large difference in the sensitivity using different fluorescent probes and
  2. The authors should redraw the figures5,7 and 9 as the study by plotting mutant/wild type across different concentrations across the different concentration H202, t-BOOH, CHP as this comparison will be much more meaningful to the readers.

Author Response

Comments and Suggestions for Authors

In this work, the authors have measured Oxidative stress in E. coli using different fluorophore in B-type wild type strain and strain lacking the arsenal to mitigate oxidative stress namely oxyR alone or oxyR along with Superoxide dismutase sodA and sodB. The authors have titrated the amount to fluorophore and oxidizing agent in such a fashion that all the experiments were conducted in the sublethal range of the oxidizing agents. Using the flow cytometric assay, the authors were able to show dose dependent response to oxidizing agents. Mito-PY1 showed the greatest sensitivity when the bacterial strains were challenged with H2O2, whereas DHDCF and HE outperformed other dyes in response to t-BOOH and CHP.

Although, the authors were able to measure reactive oxygen species using the fluorescence dye, the study failed to replicate the well-established fact in the field of oxidative stress response: deletion of sodA and sodB leads to oxidative stress response. In most cases, the authors showed there was a minor difference between in sensitivity between wild type strain and strain lacking oxyR, soda and sodB. For example, when challenged with H2O2, there was less than 2-fold change in the fluorescence intensity of Mito-PY1 between wild type and the strains lacking the superoxide dismutase. This was rather bizarre as treating the cells with H2O2 should have caused significant oxidative stress in strains lacking soda and sodB.

Response: We thank very much the reviewer for his opinion and suggestions.

The authors should clarify the following points:

Why was less oxidative stress detected in a strain lacking oxyR, sodA and sodB. It is well known in the filed of oxidative stress response that deletion of sodA and sodB leads severe growth defect due to oxidative stress that is more than 1000-fold compared to wild type. Whereas the authors have failed to any significant large difference in the sensitivity using different fluorescent probes.

Response:

The reviewer is right in that deficiency in SodA and SodB leads to increased oxidative stress in E. coli and other bacteria. This is due to the biological role of the two superoxide dismutases coded by sodA and sodB genes, i.e. the elimination of the superoxide anion via dismutation to H2O2. Eventually, H2O2 would be detoxified by catalase, yielding H2O and O2. Thus, the deficiency in sodA, sodB or both is mostly related to the intracellular accumulation of superoxide radical in aerobic conditions and, most especially, under the action of superoxide-generating agents, such as redox-cycling compounds. In contrast, catalases represent the major protection in E. coli against the consequences of peroxide-induced oxidative stress. katG and katE are the downstream catalase genes under the control of oxyR operon, which in turn is the sensor of H2O2. Therefore, oxyR or katG/katE mutants accumulate intracellular H2O2 in aerobic conditions and, most especially, under the action of exogenous peroxides, such as the ones we have used in our study. Accordingly, it is well known that only bacteria simultaneously defective in both katG and katE or sodA and sodB genes are hypersensitive with respect to mutability by peroxide and superoxide, respectively (Ruiz-Laguna, J., Prieto-Álamo, M.-J. and Pueyo, C. (2000) Oxidative mutagenesis in Escherichia coli strains lacking ROS-scavenging enzymes and/or 8-oxoguanine defenses. Environ. Mol. Mutagen. 35: 22-30. https://doi.org/10.1002/(SICI)1098-2280(2000)35:1<22::AID-EM4>3.0.CO;2-X). Since both mutant strains are deficient in oxyR (IC203: DOxyR; IC5233: DOxyRsodAB) and the prooxidants used do not generate extensively superoxide, it is not unexpected that the IC5233 strain does not overexceeds amply strain IC203 in intracellular H2O2, although in most cases (except for t-BOOH and HE probe) the levels of oxidative stress are higher in the IC5233 strain. On the contrary, strain IC5233 shows clearly higher sensititvity to superoxide-generating prooxidants, such as the redox-cycling compounds menadione, plumbagin and paraquat, when examined with the superoxide-sensitive probes HE and MitoSox Red Dye (B. Jávega, Ph.D Thesis and Manuscript in Preparation).

Regarding the severe growth defect due to oxidative stress (that would be more than 1000-fold compared to wild type, as stated by the reviewer) it has been demonstrated many years ago (Carlioz A, Touati D. Isolation of superoxide dismutase mutants in Escherichia coli: is superoxide dismutase necessary for aerobic life? EMBO J. 1986 Mar;5(3):623-30. doi: 10.1002/j.1460-2075.1986.tb04256.x. PMID: 3011417; PMCID: PMC1166808.) that a sodA sodB double mutant completely devoid of SOD was able to grow aerobically in rich medium. On the contrary, the double mutant was only unable to grow aerobically on minimal glucose medium. Since our experimental conditions involve samples studied during exponential growth in rich medium, extensive cytotoxicity of aerobic conditions to our mutant strains should not be expected. Nevertheless, our results show a slightly increased sensitivity of the triple mutant strain to the exogenous peroxides studied. We must point out that our analytical strategy implies short-time exposure to prooxidant concentrations much lower than their IC50, i.e., mild oxidative conditions, plus the fact of gating on live cells for cytometric determination of ROS. These conditions are not expected to reveal large effects on cell viability or intracellular oxidative stress and rather show the sensitivity of flow cytometric procedures, along to what has been already demonstrated previously by many groups.

The authors should redraw the figures 5,7 and 9 as the study by plotting mutant/wild type across different concentrations across the different concentration H202, t-BOOH, CHP as this comparison will be much more meaningful to the readers.

Response: Figs. 5, 7 and 9 have been redrawn according to the reviewer suggestion in order to facilitate the comparison between strains and oxidants.

Round 2

Reviewer 2 Report

Dear authors,

After carefully reading of the updated manuscript version with all the added improvements, I only can strongly encourage the acceptance of this work for further publication in the International Journal of Molecular Sciences. 

Author Response

We thank very much the reviewer for his/her opinion.